# Experimental and quantitative observational research in fencing and wheelchair fencing: A scoping review

Katharine Holmes[1], Lindsay Bottoms[2]*

1 Department of Medical Education, Icahn School of Medicine at Mount Sinai, New York, New York, United States of America, 2 Department of Psychology, Sport, and Geography, University of Hertfordshire, Hatfield, United Kingdom

* l.bottoms@herts.ac.uk

## Abstract

### Background

Fencing and wheelchair fencing are Olympic and Paralympic sports with growing global participation and increasing scientific interest. However, the overall structure, methodological profile, and thematic distribution of experimental and quantitative observational research in both has not been systematically characterized.

### Objective

To map the scope, methodological characteristics, and thematic focus of experimental and quantitative observational studies involving fencing and wheelchair fencing athletes.

### Methods

A scoping review was conducted in accordance with PRISMA-ScR guidelines. PubMed, Scopus, and EBSCOhost were searched for studies containing the terms "fencing," "fencer," or "fencers" in the title or abstract. Eligible studies employed experimental or quantitative observational designs and included fencing athletes as participants. Data were extracted using a structured framework and summarized descriptively across study design, research domain, participant characteristics, sample size, weapon discipline, and geographic distribution.

### Results

A total of 445 studies met inclusion criteria. Publication volume increased substantially after 2015. Laboratory-based (35.7%) and cross-sectional (30.3%) designs predominated, whereas prospective cohort studies (5.4%) and randomized controlled trials (4.3%) were comparatively uncommon. Performance and skill analysis constituted the largest research domain (41.1%), while injury/epidemiology (7.9%),

**Data availability statement:** The full data set is available at University of Hertfordshire Research Archive (UHRA) via DOI 10.18745/ds.00026640.

**Funding:** The author(s) received no specific funding for this work.

**Competing interests:** The authors have declared that no competing interests exist.

recovery/rehabilitation (4.3%), and training load/fatigue (1.6%) research were limited. Most studies involved small sample sizes with fewer than 50 participants and focused on able-bodied athletes; wheelchair fencing was markedly underrepresented. Sex was not specified in 24.3% of studies, and weapon discipline was not reported in 44.9%. Research output was geographically concentrated in Europe and select North American and East Asian countries.

## Conclusions

Although fencing research has expanded rapidly in recent years, it remains methodologically and thematically uneven. Greater emphasis on longitudinal and interventional designs, injury surveillance, rehabilitation research, improved reporting practices, and inclusive representation across sex, weapon discipline, geographic regions, and wheelchair athletes are necessary to strengthen the translational relevance and evidence-based development of fencing sport science.

## Introduction

Fencing and wheelchair fencing are Olympic and Paralympic combat sports characterized by high-intensity, intermittent exchanges requiring rapid acceleration, deceleration, unilateral lunging, and complex tactical decision-making [1–5]. Fencing is one of only four sports to have been included in every modern Olympic Games since 1896 [6], and wheelchair fencing has been contested at every Paralympic Games since its debut in Rome in 1960 [7]. Both disciplines require technical precision, reaction speed, neuromuscular coordination, and strategic adaptability under time pressure [8]. Although locomotor mechanics differ between able-bodied and wheelchair formats, each involves rapid upper extremity actions, dynamic balance or postural control, and repeated high-intensity efforts within structured competitive bouts.

Participation in fencing often spans from early childhood into later adulthood, resulting in a broad age range and diverse developmental pathways. International engagement has expanded substantially in recent decades across both able-bodied and wheelchair formats. The International Fencing Federation (FIE) comprises 150 member nations [9], World Para Fencing (WPF) 73 member nations [10], and USA Fencing reports over 40,000 registered members, reflecting the substantial global and national presence of the sport [11,12].

Scientific investigation has expanded alongside this growth, encompassing biomechanics, physiology, performance analysis, psychology, and injury research. Studies have examined movement mechanics [1,13,14], reaction time [2,5,15], energy system demands [8,16], tactical patterns [17,18], and musculoskeletal injury profiles [19–22]. However, the literature remains dispersed across heterogeneous methodologies and thematic domains, and its overall structural profile has not been systematically characterized. Without such mapping, it is difficult to identify methodological imbalances, thematic concentration, reporting gaps, or underrepresented athlete populations.

Scoping reviews are well suited to describe the breadth and organization of complex research fields, particularly where study designs and outcomes are diverse. Rather than evaluating intervention effects, scoping reviews provide structured overviews of study characteristics, highlight gaps, and inform future research priorities. To date, no comprehensive scoping review has mapped the experimental and quantitative observational literature in fencing. In sport science, structural concentration within certain methodologies or domains may shape practice, funding priorities, and athlete support systems. Mapping these patterns is therefore necessary not only to describe the literature but to inform strategic development of balanced, inclusive, and translationally relevant research agendas.

The objective of this study was therefore to systematically characterize the scope, methodological profile, and thematic distribution of experimental and quantitative observational research involving fencing athletes. By examining publication trends, study designs, research domains, participant characteristics, and geographic representation, this review aims to provide a structural overview of the field and support more balanced and inclusive development of fencing sport science.

## Materials and methods

### Study design

This study was conducted as a scoping review in accordance with the Preferred Reporting Items for Systematic Reviews and Meta-Analyses Extension for Scoping Reviews (PRISMA-ScR) guidelines. The review protocol was prospectively registered on the Open Science Framework (OSF) to enhance transparency and methodological rigor. The objective of this review was to systematically map the scope, methodological structure, and thematic distribution of experimental and quantitative observational research involving fencing athletes. The guiding research question was: What is the scope, methodological profile, and thematic focus of experimental and quantitative observational research conducted in fencing athletes?

### Eligibility criteria

Studies were eligible for inclusion if they employed an experimental or quantitative observational design and included fencing athletes as study participants. Eligible designs included randomized controlled trials, non-randomized interventions, controlled laboratory experiments, field-based experiments or testing, prospective cohort studies, retrospective observational studies, and cross-sectional studies with quantitative outcomes. Studies involving multiple sports were included only if fencing athletes were analyzed as a distinct subgroup and fencing-specific results were reported. Eligible outcomes included any quantitative measures related to injury, biomechanics, physiology, performance, training load, fatigue, recovery, health, or psychological factors. Studies were excluded if they were case reports, case series, qualitative-only studies, narrative reviews, editorials, coaching commentaries, historical or rule-based articles, or if fencing participants were aggregated with other sports without fencing-specific analyses. No restrictions were placed on publication date or language.

### Search strategy

A comprehensive literature search was conducted across PubMed, Scopus, and EBSCOhost on November 18, 2025. The search strategy intentionally used the broad terms "fencing," "fencer," and "fencers" in the title or abstract to maximize sensitivity while maintaining specificity for fencing-related literature. The search strings were as follows: PubMed: ("fencing"[Title/Abstract] OR "fencer"[Title/Abstract] OR "fencers"[Title/Abstract]); Scopus: TITLE-ABS ("fencing" OR "fencer" OR "fencers"); and EBSCOhost: TI fencing OR TI fencer OR TI fencers OR AB fencing OR AB fencer OR AB fencers. Pilot testing of alternative search approaches suggested this strategy captured the broadest and most relevant body of literature across databases. This approach was also informed by the authors' familiarity with the fencing literature and was considered sufficient to identify the major body of eligible studies. However, it is possible that studies referring only

to specific disciplines (e.g., foil or sabre) without explicit reference to fencing in the title or abstract may not have been captured, which represents a potential limitation. All records were exported and imported into Rayyan (Rayyan Systems Inc., 2025) for deduplication and screening, a platform developed to support efficient and reproducible study screening in evidence synthesis [23]. The initial search yielded 9,770 records. After removal of 4,180 duplicate records, 5,590 unique records remained for screening.

## Study selection

Screening was conducted in a multi-stage process. Titles were first reviewed to exclude clearly irrelevant studies, followed by abstract screening. Full-text review was performed when eligibility could not be determined from the abstract alone. Full-text articles were sought for all potentially eligible studies. When full texts could not be obtained through institutional subscriptions, interlibrary loan, or direct library inquiry, studies were excluded. A total of 57 records were excluded due to inability to obtain the full text. Screening and study selection were conducted by a single reviewer using predefined eligibility criteria. Given the descriptive and mapping focus of scoping reviews, single-reviewer screening was deemed acceptable. For studies in which eligibility could not be determined based on title, abstract, or full-text review, an AI-assisted data extraction tool (Elicit; Ought, Inc., 2026) was used to facilitate clarification of key inclusion criteria: (1) whether fencers were included as study participants; (2) whether the study employed an experimental or quantitative observational design; and (3) for studies involving multiple sports, whether results for fencing athletes were reported separately. AI-assisted outputs were used solely to support identification of relevant content and did not make or replace inclusion or exclusion decisions. All outputs were manually reviewed and verified by the reviewer, and final eligibility determinations were made according to predefined criteria. Non-English language studies were screened and assessed using translated abstracts and full texts generated using an AI-assisted language model to support comprehension and inclusion, with translated outputs reviewed in context of the original article structure and used only to support screening and extraction.

## Data extraction

Data were charted using a structured extraction framework developed a priori. Extracted variables included publication year, first author country, sport type, athlete type, number of fencing participants, fencer level, weapon discipline, sex of participants, primary research domain, study design, primary outcome, and key findings. To facilitate structured extraction across the large number of included studies (n = 445), an AI-assisted extraction platform was used with standardized categorical fields applied consistently across batches. Extracted data were manually reviewed, cross-checked against source manuscripts, and standardized prior to descriptive synthesis. To support consistency and accuracy, the same extraction framework and categorical definitions were applied across studies, and any uncertainties or discrepancies identified during review were resolved by the reviewer before final coding. This reviewer-led verification approach is consistent with emerging evidence suggesting Elicit may support semi-automated extraction workflows but benefits from human oversight to support accuracy and validity [24].

## Data synthesis

Findings were summarized using descriptive numerical analyses and presented in tabular and graphical formats. Frequency distributions and percentages were calculated for study design, research domain, participant characteristics, weapon discipline, sample size categories, and geographic distribution. Narrative synthesis was used to contextualize structural patterns across the literature. Consistent with scoping review methodology, no formal risk-of-bias assessment was performed.

This study involved analysis of published literature and did not require institutional review board approval.

## Results

### Study selection

The literature search identified 9,770 records across PubMed, Scopus, and EBSCOhost. After removal of 4,180 duplicate records, 5,590 unique records remained for screening. Following title and abstract review and full-text assessment, 57 studies were excluded due to inability to obtain the full text despite attempts through institutional access and interlibrary loan. A total of 445 studies met eligibility criteria and were included in the final synthesis. The study selection process is illustrated in Fig 1.

### Characteristics of included studies

**Publication trends.** Publication volume increased progressively over time (Fig 2). Prior to 2000, relatively few studies were published annually, with most years yielding fewer than five publications. Output increased gradually in the early 2000s and accelerated markedly after 2015. Annual publication counts exceeded 20 studies per year after 2018 and peaked in the early 2020s, approaching 40 studies in a single year. The majority of included studies were published within the past decade.

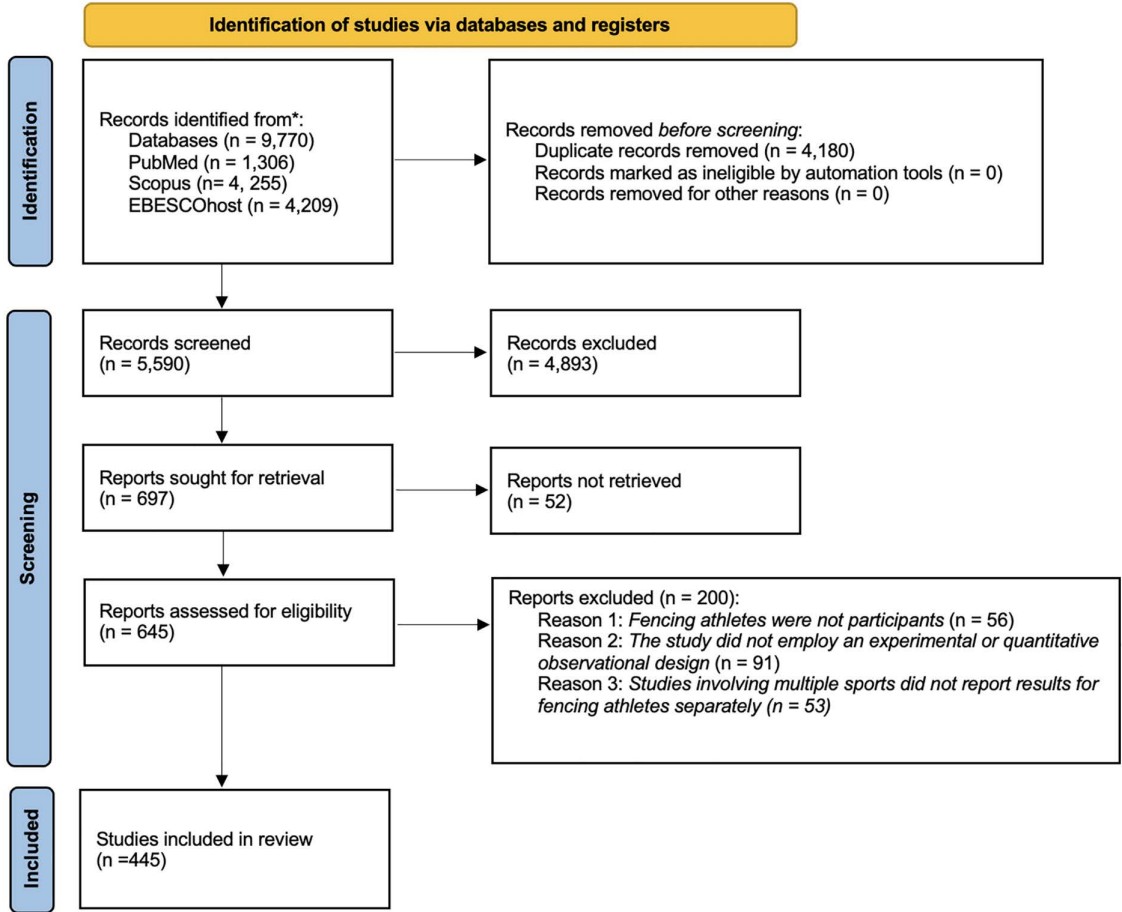

**Fig 1. PRISMA-ScR flow diagram of study identification, screening, eligibility, and inclusion.** Flow diagram illustrating the process of study identification, screening, eligibility assessment, and inclusion. A total of 9,770 records were identified across PubMed, Scopus, and EBSCOhost. After removal of 4,180 duplicates, 5,590 records were screened. Following full-text assessment and application of eligibility criteria, 445 studies were included in the final synthesis.

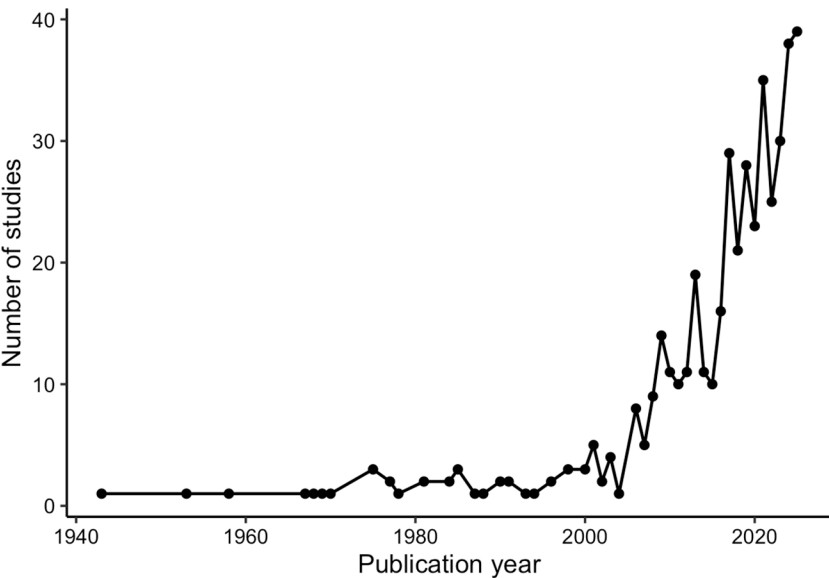

**Fig 2. Annual number of included studies published per year.** Line graph depicting the number of included studies published per year. Publication volume increased progressively over time, with marked growth after 2015 and the highest annual counts observed in the early 2020s.

**Study design.** Controlled laboratory experiments were the most frequently represented design (35.7%), followed by cross-sectional observational studies (30.3%) and field-based experimental or testing studies (12.1%) (Table 1). Retrospective observational studies comprised 7.6% of included studies, prospective cohort studies 5.4%, randomized controlled trials 4.3%, and non-randomized interventions 2.5%. A small proportion of studies were categorized as other designs (1.6%) or involved multiple study designs (0.4%). Overall, laboratory-based and cross-sectional approaches accounted for the majority of the literature, whereas prospective and interventional designs were comparatively limited.

**Research domain.** Performance and skill analysis was the dominant research domain, representing 41.1% of included studies (Table 2). Physiology accounted for 15.7%, psychology 13.0%, and biomechanics 12.6%. Injury and epidemiology studies comprised 7.9% of the literature, while recovery, rehabilitation, and health-focused research represented 4.3%. Training load and fatigue studies were least represented at 1.6%. A small proportion of studies (2.9%) addressed overlapping domains, and 0.9% were categorized as Other. Together, performance, physiology, psychology, and biomechanics accounted for 82.4% of the literature, highlighting concentration within performance-oriented domains.

## Participant characteristics

Most studies examined able-bodied athletes (95.5%), whereas wheelchair athletes accounted for 3.4% of included studies and 0.9% included both populations (Table 3). Athlete type was unspecified in 0.2% of studies. With respect to sex, 44.6% of studies included both male and female participants, 20.9% included males only, and 10.3% included females only. Sex was not specified in 24.3% of studies. Senior or elite fencers were the most frequently studied population (54.4%). Mixed-level cohorts accounted for 12.8%, while youth and collegiate populations comprised 8.5% and 8.3% of studies, respectively. Junior fencers represented 4.0% of studies, and club-level athletes accounted for 2.7%. Veteran fencers were rarely studied (0.2%), and 9.0% of studies did not specify competition level. Weapon discipline reporting remained inconsistent. Although 22.0% of studies examined multiple weapons, 16.9% focused on épée and 12.4% on foil. Sabre accounted for 3.8% of studies, while 44.9% of studies did not specify weapon discipline.

**Table 1. Distribution of study designs across included studies (n = 445).**

| Study Design | N (%) |
|---|---|
| Controlled Laboratory Experiment | 159 (35.7) |
| Cross-Sectional Observational | 135 (30.3) |
| Field-Based Experiment/Testing | 54 (12.1) |
| Retrospective Observational | 34 (7.6) |
| Prospective Cohort | 24 (5.4) |
| Randomized Controlled Trial (RCT) | 19 (4.3) |
| Non-Randomized Intervention | 11 (2.5) |
| Other | 7 (1.6) |
| Multiple Designs | 2 (0.4) |

**Table 2. Distribution of research domains across included studies (n = 445).**

| Research Domain | N (%) |
|---|---|
| Performance/Skill Analysis | 183 (41.1) |
| Physiology | 70 (15.7) |
| Psychology | 58 (13.0) |
| Biomechanics | 56 (12.6) |
| Injury/Epidemiology | 35 (7.9) |
| Recovery/Rehab/Health | 19 (4.3) |
| Overlapping Domains | 13 (2.9) |
| Training Load/Fatigue | 7 (1.6) |
| Other | 4 (0.9) |

**Sample size.** Sample sizes were generally small (Fig 3). Most studies included fewer than 50 participants, with the 10–19 and 20–49 participant ranges being most common. A substantial proportion of studies included fewer than 10 athletes. Only a small minority of studies enrolled 100 or more participants.

## Geographic distribution

Research output was geographically concentrated in Europe and North America (Table 4). Poland contributed the largest proportion of publications (11.2%), followed by the United States (8.3%), France (8.1%), Italy (6.7%), and the United Kingdom (6.3%). East Asian countries including China, Japan, and South Korea were also represented. Contributions from Africa and South America were comparatively limited, and many countries contributed only one or two publications.

## Discussion

### Principal findings

This scoping review synthesized 445 experimental and quantitative observational studies involving fencing athletes. Publication volume has increased substantially over the past decade, with marked acceleration after 2015. Despite this growth, the literature demonstrates consistent structural patterns: predominance of laboratory-based and cross-sectional designs, concentration within performance and physiology domains, frequent small sample sizes, limited representation of wheelchair athletes, and geographic clustering within Europe and North America. Together, these findings suggest that while fencing research has expanded in quantity, its methodological structure and thematic distribution remain uneven.

**Table 3. Participant characteristics across included studies (N = 445).**

| | N (%) |
|---|---|
| **Sex of Participants** | |
| Both | 198 (44.6) |
| Not specified | 108 (24.3) |
| Males only | 93 (20.9) |
| Females only | 46 (10.3) |
| **Athlete Type** | |
| Able-bodied | 425 (95.5) |
| Wheelchair | 15 (3.4) |
| Both | 4 (0.9) |
| Not specified | 1 (0.2) |
| **Weapon Studied** | |
| Not specified | 200 (44.9) |
| Multiple | 98 (22.0) |
| Épée | 75 (16.9) |
| Foil | 55 (12.4) |
| Sabre | 17 (3.8) |
| **Fencer Level** | |
| Senior Elite | 242 (54.4) |
| Mixed levels | 57 (12.8) |
| Not Specified | 40 (9.0) |
| Youth | 38 (8.5) |
| Collegiate | 37 (8.3) |
| Junior | 18 (4.0) |
| Club | 12 (2.7) |
| Veteran | 1 (0.2) |

Collectively, these findings suggest a structural imbalance within fencing research characterized by: 1) methodological concentration in laboratory and cross-sectional designs; 2) thematic concentration in performance-oriented domains, and 3) limited representation of para-athletes, longitudinal cohorts, and rehabilitation-focused inquiry. This pattern appears to reflect a greater emphasis within the literature on performance characterization than athlete health surveillance and intervention evaluation. While such emphasis has advanced understanding of fencing-specific motor and perceptual demands, it may also constrain the translation of findings into long-term athlete development, injury prevention, and inclusive practice frameworks.

## Methodological profile of the field

Controlled laboratory experiments and cross-sectional observational studies comprised the majority of included research, whereas prospective cohort studies and randomized controlled trials were comparatively uncommon. This pattern suggests that fencing research has been structured primarily around performance testing and biomechanical or physiological assessment rather than longitudinal surveillance or intervention-based investigation.

Fencing participation often spans from early childhood into later adulthood, underscoring the importance of longitudinal designs capable of capturing developmental trajectories, cumulative training exposures, injury patterns, and age-related performance changes. The relative scarcity of prospective cohort studies limits the ability to establish temporal relationships between training exposure and outcomes such as overuse injury, performance adaptation, or burnout. Without large-scale surveillance systems, incidence rates and modifiable risk factors remain difficult to define. Collaborative

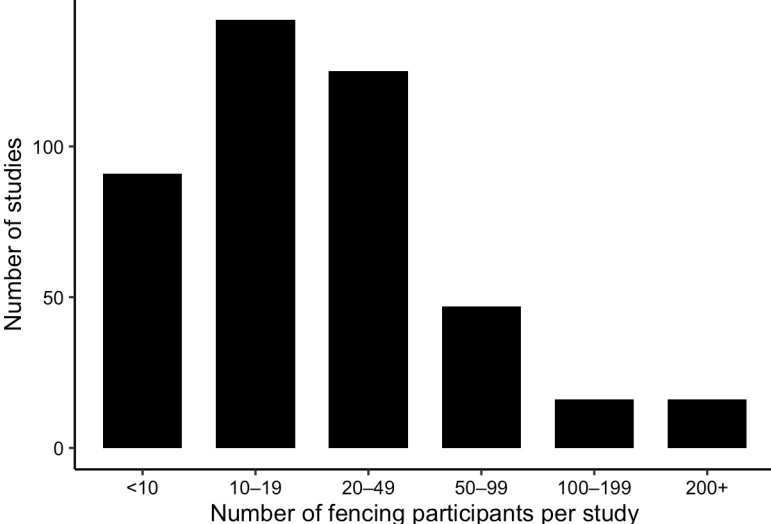

**Fig 3. Distribution of fencing participant sample sizes across included studies.** Bar chart showing the distribution of sample sizes among included studies (n = 445), grouped into predefined participant ranges (<10, 10–19, 20–49, 50–99, 100–199, and ≥200 participants). Most studies included fewer than 50 fencing participants.

**Table 4. Geographic distribution of included studies by first author country (top 10).**

| Country | N (%) |
|---|---|
| Poland | 50 (11.2) |
| United States | 37 (8.3) |
| France | 36 (8.1) |
| Italy | 30 (6.7) |
| United Kingdom | 28 (6.3) |
| Spain | 25 (5.6) |
| China | 19 (4.3) |
| Japan | 16 (3.6) |
| Ukraine | 16 (3.6) |
| South Korea | 16 (3.6) |

multi-center studies and federation-level data collection initiatives may therefore represent important priorities for advancing long-term athlete health research in fencing.

Similarly, the limited number of randomized or controlled intervention studies constrains the field's ability to move beyond descriptive findings toward evidence-based practice. Intervention trials are necessary to evaluate strategies aimed at reducing injury risk, optimizing training load, improving rehabilitation protocols, and enhancing performance. In their absence, recommendations may rely primarily on cross-sectional associations or laboratory findings rather than causal evidence.

The predominance of small sample sizes likely reflects the logistical challenges inherent in recruiting fencing athletes, particularly at senior and elite levels. Although fencing participation is expanding in several countries, the overall athlete pool remains relatively small compared with larger team sports, and elite fencers are often concentrated within a limited number of centralized training environments. This structural reality constrains recruitment capacity for single-site investigations. Nevertheless, small samples and short-duration designs reduce statistical power and may limit generalizability across weapon disciplines, age groups, and competition levels. They also restrict subgroup analyses, including

sex-specific and developmental comparisons, which are particularly relevant in a sport characterized by early specialization and prolonged competitive careers. Multi-centre collaboration and federation-level research partnerships may therefore represent important strategies for strengthening statistical robustness while maintaining ecological validity.

## Thematic concentration and research gaps

Performance and skill analysis constituted the largest research domain, representing over 40% of the literature. In contrast, injury and epidemiology studies accounted for less than 10%, and recovery or rehabilitation-focused research represented fewer than 5%. Training load and fatigue research was particularly limited. The near absence of structured training load and fatigue monitoring research is particularly notable given fencing's intermittent high-intensity profile and asymmetrical mechanical demands.

Fencing involves repetitive unilateral loading, high-intensity lunging mechanics, and asymmetrical lower extremity demands, all of which have been associated with lower limb injury patterns in prior studies [11,19–22,25–29]. Comparatively fewer investigations have examined upper extremity injuries, despite the substantial technical demands placed on the weapon arm and shoulder complex [24,30–34]. The limited emphasis on comprehensive and longitudinal injury surveillance may result in greater capture of acute injuries than overuse conditions, despite the importance of both in fencing. Greater investment in prospective injury tracking, load monitoring, and rehabilitation-focused research may therefore be necessary to better characterize risk patterns and inform prevention strategies that support long-term athlete participation.

Sex representation and reporting practices also warrant careful consideration. Nearly one quarter of studies did not specify participant sex, and fewer than 11% included female-only cohorts. Although approximately half of studies included both male and female participants, inconsistent reporting limits meaningful sex-specific interpretation. This limitation is particularly consequential given established biological and physiological differences between females and males including variations in body composition, hormonal milieu, and neuromuscular characteristics that may influence injury patterns, training adaptation, and recovery trajectories [35,36]. Moreover, international surveillance data indicate that injury incidence and vulnerability may differ by sex in fencing populations [32,37] further underscoring the importance of sex-disaggregated analysis. Female fencers represent a substantial and growing proportion of the sport; approximately 40% of USA Fencing membership identifies as female, comparable to participation levels within the National Collegiate Athletic Association (~43%) [38]. Despite this representation, sex-specific research in fencing remains limited. Addressing these differences is critical not only for sport-specific performance and injury prevention but also for the broader long-term health of female athletes.

The dominance of performance and skill analysis likely reflects several structural features of fencing sport science. As an Olympic and Paralympic discipline, fencing research is often embedded within high-performance environments where funding, institutional support, and practitioner interest may emphasize competitive optimization. Performance variables such as reaction time, movement velocity, perceptual-cognitive processing, and tactical decision-making are also well suited to controlled laboratory paradigms, enabling relatively efficient data collection within small cohorts. In contrast, injury surveillance, longitudinal load monitoring, and rehabilitation research require sustained multi-site collaboration, long-term follow-up, and integration with medical infrastructures. Consequently, methodological feasibility and performance-oriented funding structures may have contributed to shaping the thematic distribution of the literature.

## Representation and inclusivity

Wheelchair fencing was markedly underrepresented in the included literature, comprising only a small fraction of studies. This imbalance is notable given its longstanding status as a Paralympic discipline and its global reach. Wheelchair fencing has been contested at every Paralympic Games since its debut in Rome in 1960 and has expanded substantially in scope and participation [7]. Despite this history, systematic research in wheelchair fencing remains very limited, with few studies examining the physical, physiological, and technical characteristics of para-athletes [39]. This underrepresentation mirrors

broader disparities in Paralympic sport research and highlights the need for equitable scientific investment. Considering the unique postural, neuromuscular, and tactical demands of wheelchair fencing, greater research investment is needed to support evidence-based training, injury prevention, rehabilitation, and performance development for para-athletes.

The literature was also disproportionately focused on senior or elite fencers, despite these athletes representing only a small fraction of the overall fencing population. While elite cohorts provide valuable insight into high-performance demands, this emphasis may limit generalizability to youth, collegiate, recreational, and veteran athletes who comprise the majority of participants globally. Broader inclusion of non-elite populations is necessary to better characterize developmental trajectories, injury patterns, and long-term participation across the full spectrum of fencing engagement.

Similarly, weapon discipline was not specified in nearly half of included studies, limiting discipline-specific interpretation. Relatively few investigations have directly compared physiological or injury differences across épée, foil, and sabre, despite evidence that meaningful distinctions exist. Injury risk appears to vary by weapon, with international data suggesting differences in overall injury incidence and sex-specific vulnerability across disciplines [33–37]. Mechanisms of injury also differ, with a substantial proportion arising from intrinsic, non-contact efforts [37] and sabre presenting unique risk patterns that have historically prompted equipment standard modifications [31]. These variations likely reflect distinct tactical and biomechanical demands across weapons. Sabre is characterized by rapid, explosive exchanges, whereas épée involves longer, more sustained tactical actions, with foil occupying an intermediate profile shaped by right-of-way conventions and structured engagement patterns [12,40]. Such temporal and tactical distinctions likely translate into different physiological demands, neuromuscular loading profiles, and injury risk patterns. Given these discipline-specific characteristics, failure to consistently report or stratify by weapon may obscure meaningful differences in performance outcomes, adaptation, and injury risk. Improved reporting practices and weapon-specific analyses would enhance comparability across studies and strengthen the applicability of findings to training, prevention, and rehabilitation strategies.

Geographically, research output was concentrated in European nations and select North American and East Asian countries. While this distribution reflects established fencing infrastructures and sport science investment, limited representation from other regions may constrain generalizability to global fencing populations. National federation structures, centralized coaching models, and differences in access to sport science and medical resources may influence athlete development pathways, training load management, and rehabilitation practices. Broader geographic diversity in research participation would strengthen external validity and enhance understanding of contextual influences on athlete health and performance.

## Implications for future research

This structural mapping suggests several priorities for future investigation. Increased use of prospective cohort designs and randomized interventions may strengthen causal inference and translational relevance. Expanded injury surveillance and rehabilitation-focused research may better support long-term athlete health. Improved reporting of sex, weapon discipline, and competition level would enhance transparency and comparability. Finally, greater inclusion of wheelchair fencing and broader international collaboration may promote equity and global relevance within the field.

## Research gaps and strategic priorities

The structural imbalances identified in this review suggest several strategic priorities for advancing fencing sport science. These gaps are methodological, thematic, and representational in nature, and collectively shape the field's translational capacity. To consolidate these priorities, key research gaps and recommended directions are summarized in Table 5.

Collectively, these priorities suggest that the next phase of fencing research should move toward greater methodological diversity, interdisciplinary integration, and collaborative scale. Addressing these gaps may benefit from coordinated partnerships between academic institutions, national federations, high-performance programs, and sports medicine infrastructures. Such collaboration may enable larger sample sizes, longitudinal surveillance systems, and intervention-based research capable of informing evidence-based practice across developmental stages and competition levels.

**Table 5. Key research gaps and priority directions for fencing sport science.**

| Domain | Identified Gap | Why It Matters | Priority Research Direction |
|---|---|---|---|
| **Study Design** | Limited prospective cohort studies; few randomized or controlled interventions | Restricts causal inference, understanding of injury incidence, training adaptation, and developmental trajectories | Establish multi-center longitudinal surveillance systems and conduct pragmatic or cluster-based intervention trials |
| **Training Load & Fatigue** | Minimal structured load monitoring research | Inhibits understanding of load–injury and load–performance relationships | Develop fencing-specific load metrics, monitoring tools, and periodization models |
| **Injury & Rehabilitation** | Limited prospective injury tracking; scarce rehabilitation-focused research | Constrains injury prevention strategies and return-to-play guidance | Integrate sport science and medical data to support injury risk modelling and rehabilitation protocol evaluation |
| **Sex Representation** | Inconsistent sex reporting; few female-only cohorts | Limits sex-specific interpretation of injury patterns, adaptation, and performance | Mandate sex-disaggregated analyses and expand female-focused research initiatives |
| **Weapon Discipline** | Nearly half of studies do not specify weapon discipline | Obscures discipline-specific physiological and biomechanical demands | Standardize weapon reporting and conduct comparative weapon-specific investigations |
| **Para Sport** | Wheelchair fencing markedly underrepresented | Limits equitable evidence-based training, injury prevention, and rehabilitation support | Develop para-specific biomechanical, physiological, and performance research programs |
| **Population Scope** | Heavy emphasis on elite athletes | Limits generalizability to youth, recreational, collegiate, and veteran fencers | Expand research across developmental stages and broader participation levels |
| **Geographic Representation** | Concentration of research in Europe and North America | Limits global applicability and contextual understanding | Foster international and federation-level research collaborations |
| **Reporting Standards** | Inconsistent reporting of sex, weapon discipline, and competition level | Reduces reproducibility, comparability, and synthesis capacity | Develop fencing-specific reporting recommendations aligned with sport science best practices |
| **Interdisciplinary Integration** | Domains frequently studied in isolation | Reduces holistic understanding of performance, health, and long-term development | Promote integrated multi-domain research frameworks combining biomechanics, physiology, psychology, and medical surveillance |

## Strengths and limitations

This review synthesized a large body of experimental and quantitative observational research involving fencing athletes using a comprehensive multi-database search strategy without restriction on publication date or language. Screening and reporting were conducted in accordance with PRISMA-ScR guidelines, and data were extracted using a structured framework developed a priori. Inclusion of non-English studies enhances the global scope of the review and reduces language bias. The large number of included studies (n = 445) provides a broad structural overview of the field across domains, methodologies, and geographic regions.

Several limitations should be acknowledged. Screening and study selection were conducted by a single reviewer, and independent dual screening was not performed. This may have introduced the possibility of missed studies or inconsistent application of eligibility criteria. However, consistent with the descriptive and mapping objectives of scoping review methodology, a single-reviewer approach was deemed acceptable for this review. AI-assisted tools were used to support screening and data extraction by identifying relevant manuscript sections; however, final eligibility decisions and categorizations were determined by the reviewer. As a scoping review, this study did not assess methodological quality or risk of bias within individual studies. Additionally, the substantial heterogeneity in study designs, outcomes, and domains limited synthesis to descriptive mapping rather than quantitative aggregation or meta-analysis.

## Conclusions

Fencing research has expanded substantially over the past decade, with growing publication volume and increasing international contribution. However, the current evidence base remains structurally concentrated within laboratory-based and cross-sectional designs and thematically centered on performance and physiological domains. Prospective cohort

studies, randomized interventions, injury surveillance, rehabilitation research, and inclusive representation of wheelchair athletes remain comparatively limited. Inconsistent reporting of key variables— including sex, weapon discipline, and competition level—further constrains interpretability and comparability across studies. Greater methodological diversity, improved reporting practices, expanded longitudinal surveillance, and broader geographic and para-athlete inclusion may strengthen the translational relevance and global applicability of fencing research. By systematically mapping the methodological and thematic structure of the field, this review provides a foundation for targeted advancement of fencing sport science and supports more balanced development of research priorities moving forward.

## Supporting information

**S1 Checklist. PRISMA-ScR Checklist.** Completed Preferred Reporting Items for Systematic reviews and Meta-Analyses extension for Scoping Reviews checklist for the scoping review.
(DOCX)

## Author contributions

**Conceptualization:** Katharine Holmes, Lindsay Bottoms.

**Formal analysis:** Katharine Holmes, Lindsay Bottoms.

**Methodology:** Katharine Holmes.

**Writing – original draft:** Katharine Holmes, Lindsay Bottoms.

**Writing – review & editing:** Katharine Holmes, Lindsay Bottoms.

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
