## [Decision Letter · Decision Letter 0]

19 Apr 2026

PONE-D-26-11865Experimental and Quantitative Observational Research in Fencing and Wheelchair Fencing: A Scoping ReviewPLOS One

Dear Dr. Bottoms,

Thank you for submitting your manuscript to PLOS ONE. After careful consideration, we feel that it has merit but does not fully meet PLOS ONE’s publication criteria as it currently stands. Therefore, we invite you to submit a revised version of the manuscript that addresses the points raised during the review process.

We look forward to receiving your revised manuscript.

Kind regards,

Yih-Kuen Jan, PhD

Academic Editor

PLOS One

Journal Requirements:

“This research received no specific grant from any funding agency in the public, commercial, or not-for-profit sectors.”

Reviewers' comments:

Reviewer's Responses to Questions

**Comments to the Author**

1. Is the manuscript technically sound, and do the data support the conclusions?

Reviewer #1: Yes

Reviewer #2: Yes

2. Has the statistical analysis been performed appropriately and rigorously? 

Reviewer #1: Yes

Reviewer #2: N/A

3. Have the authors made all data underlying the findings in their manuscript fully available?

Reviewer #1: No

Reviewer #2: Yes

4. Is the manuscript presented in an intelligible fashion and written in standard English?

Reviewer #1: Yes

Reviewer #2: Yes

5. Review Comments to the Author

Reviewer #1: This manuscript presents a scoping review of experimental and quantitative observational research in fencing and wheelchair fencing. The topic is timely and relevant, and the review offers a useful overview of publication trends, study designs, research domains, participant characteristics, and geographic distribution. The inclusion of 445 studies suggests substantial effort, and the manuscript is generally well written and logically structured. The broad conclusion that the literature has expanded but remains uneven in methodology and thematic coverage is reasonable and supported by the descriptive results. However, several points should be addressed to strengthen the rigor and transparency of the review.

First, the manuscript would benefit from greater methodological detail regarding study selection and data extraction. Screening and study selection were conducted by a single reviewer, which should be acknowledged more clearly as a limitation. While this may be acceptable in some scoping review contexts, it still raises the possibility of missed studies or inconsistent application of eligibility criteria.

Second, the use of AI-assisted tools in screening, extraction, and translation should be described in more detail. The current wording is helpful, but the manuscript should clarify exactly how these tools were used, how their outputs were verified, and what safeguards were in place to ensure consistency and accuracy. This is especially important because these tools were involved in several stages of the review process.

Third, the search strategy appears relatively narrow as currently described. The manuscript states that the search used the terms “fencing,” “fencer,” or “fencers” in the title or abstract across PubMed, Scopus, and EBSCOhost. The authors should provide the full search strings for each database and explain why this strategy was considered sufficiently sensitive. It would also be helpful to discuss whether potentially relevant studies using terms such as foil, sabre, or wheelchair fencing without “fencing” in the title/abstract may have been missed.

Fourth, the data availability statement should be improved. Because this review generated a substantial extracted dataset across 445 studies, the underlying extraction sheet and coding framework should ideally be made available in a public repository or as supporting information, in line with PLOS ONE data availability expectations. The current statement that data will be available on acceptance is not fully sufficient as written.

Fifth, I noticed what appears to be a presentation error in the figure captions. The captions/descriptions for Figures 2 and 3 seem to be reversed: the text under Figure 2 describes sample size distribution, while the text under Figure 3 describes annual publication counts. Please check and correct this.

I also encourage the authors to slightly moderate some of the wording in the Discussion. For example, the “Structural Imbalance Model” is an interesting interpretive summary, but it may be better presented as a conceptual interpretation of the mapped literature rather than as a formal model. Similarly, some claims about what the field has “prioritized” could be framed more cautiously.

Overall, this is a useful manuscript with clear potential value to the field. With revision to improve transparency, reproducibility, and reporting detail, it would be considerably strengthened.

Reviewer #2: This manuscript presents a scoping review aiming to characterize the methodological structure and thematic distribution of experimental and quantitative observational research in fencing and wheelchair fencing. The topic is relevant and timely, given the growing interest in niche sports science domains and the clear lack of synthesis in fencing research. The inclusion of 445 studies and adherence to PRISMA-ScR guidelines are notable strengths. However, several reporting issues can be fixed to make it better.

The author mentioned usage of AI tool in data screening and extraction, but details about the AI are insufficient and lead to less transparency and reproducibility. For example, are there any validity test done on the AI tool and what prompt was given to this tool to minimize the bias created by AI? Are there any criteria used to ensure accuracy?

Line 151: missing period.

For result section the depth of data synthesis and analysis are very limited. Most analysis was descriptive (frequencies and percentages). Deeper insight with some exploration of relationships between variables is recommended (e.g., study design × domain, geography × design).

The manuscript has clear potential and addresses an underexplored area. However, methodological transparency and analytical depth need to be improved before it is suitable for publication.

6. PLOS authors have the option to publish the peer review history of their article (what does this mean?). If published, this will include your full peer review and any attached files.

Reviewer #1: No

Reviewer #2: No

---

## [Author Response · Author response to Decision Letter 1]

21 Apr 2026

We thank the reviewers for their thoughtful critique of our manuscript and believe their comments have made the manuscript stronger and more clinically meaningful. All changes have been made in text with lines and page numbers included below.

Editor

Comment 1: Please ensure that your manuscript meets PLOS ONE's style requirements, including those for file naming.

RESPONSE: Thank you for this note. We have reviewed the manuscript and associated submission files to ensure compliance with PLOS ONE style requirements, including formatting and file naming conventions, and have revised the submission materials accordingly.

Comment 2: Thank you for stating the following in the Acknowledgments Section of your manuscript: “This research received no specific grant from any funding agency in the public, commercial, or not-for-profit sectors.” We note that you have provided funding information that is not currently declared in your Funding Statement. However, funding information should not appear in the Acknowledgments section or other areas of your manuscript. We will only publish funding information present in the Funding Statement section of the online submission form. Please remove any funding-related text from the manuscript and let us know how you would like to update your Funding Statement. Currently, your Funding Statement reads as follows: “The author(s) received no specific funding for this work.” Please include your amended statements within your cover letter; we will change the online submission form on your behalf.

RESPONSE: Thank you for pointing this out. Any reference to funding has been removed from the manuscript. The funding statement online is correct and can stay as is. The cover letter also reflects that no funding was received.

Comment 3: When completing the data availability statement of the submission form, you indicated that you will make your data available on acceptance. We strongly recommend all authors decide on a data sharing plan before acceptance, as the process can be lengthy and hold up publication timelines. Please note that, though access restrictions are acceptable now, your entire data will need to be made freely accessible if your manuscript is accepted for publication. This policy applies to all data except where public deposition would breach compliance with the protocol approved by your research ethics board. If you are unable to adhere to our open data policy, please kindly revise your statement to explain your reasoning and we will seek the editor's input on an exemption. Please be assured that, once you have provided your new statement, the assessment of your exemption will not hold up the peer review process.

RESPONSE: The data is now available online and can be found here: https://uhra.herts.ac.uk/id/eprint/26640/

Reviewer 1

Comment 1: First, the manuscript would benefit from greater methodological detail regarding study selection and data extraction. Screening and study selection were conducted by a single reviewer, which should be acknowledged more clearly as a limitation. While this may be acceptable in some scoping review contexts, it still raises the possibility of missed studies or inconsistent application of eligibility criteria.

RESPONSE: We thank the reviewer for this important comment. In response, we revised the Methods and Limitations sections to address both methodological transparency and acknowledgment of limitations.

Specifically, we expanded the Study Selection and Data Extraction sections to provide additional detail regarding screening procedures, eligibility clarification, the use and verification of AI-assisted tools, and safeguards used to support consistency and accuracy during extraction and coding. (Please see lines 117-130 and 143-152)

We also more explicitly acknowledge single-reviewer screening as a limitation and note the potential for missed studies or inconsistent application of eligibility criteria. While a single-reviewer approach was considered acceptable given the descriptive and mapping objectives of scoping review methodology, we agree that clearer acknowledgment of this limitation strengthens the manuscript. (Please see lines 441-444)

Comment 2: Second, the use of AI-assisted tools in screening, extraction, and translation should be described in more detail. The current wording is helpful, but the manuscript should clarify exactly how these tools were used, how their outputs were verified, and what safeguards were in place to ensure consistency and accuracy. This is especially important because these tools were involved in several stages of the review process.

RESPONSE: We thank the reviewer for this important comment. In response, we substantially expanded the Methods section to provide additional detail regarding the role of AI-assisted tools in screening, extraction, and translation, including how these tools were used, how outputs were verified, and what safeguards were in place to support consistency and accuracy. Specifically, we clarify that AI-assisted tools were used only to support identification and organization of relevant information and did not make autonomous inclusion, exclusion, or coding decisions. We added detail regarding the predefined criteria used when AI assistance was applied, including eligibility clarification related to participant inclusion, study design, and fencing-specific reporting in multi-sport studies. We also expanded the description of reviewer-led safeguards, including manual review and verification of all AI-assisted outputs, cross-checking against source manuscripts, use of standardized extraction fields and categorical definitions across studies, and resolution of uncertainties or discrepancies prior to final coding. In addition, we added citations supporting the screening platform used in this review (Rayyan; Ouzzani et al.) and emerging evidence supporting human oversight in Elicit-assisted extraction workflows (Hilkenmeier et al., 2025). We believe these revisions substantially improve transparency, reproducibility, and justification of the AI-assisted methods used in this review. (Please see lines 117-130 and 143-152).

Comment 3: Third, the search strategy appears relatively narrow as currently described. The manuscript states that the search used the terms “fencing,” “fencer,” or “fencers” in the title or abstract across PubMed, Scopus, and EBSCOhost. The authors should provide the full search strings for each database and explain why this strategy was considered sufficiently sensitive. It would also be helpful to discuss whether potentially relevant studies using terms such as foil, sabre, or wheelchair fencing without “fencing” in the title/abstract may have been missed.

RESPONSE: We thank the reviewer for this important comment. In response, we revised the Search Strategy section to provide the full search strings used for PubMed, Scopus, and EBSCOhost and to clarify the rationale for the search approach. Specifically, the terms “fencing,” “fencer,” and “fencers” were intentionally selected as broad search terms to maximize sensitivity while maintaining specificity for fencing-related literature. We added that pilot testing of alternative search approaches suggested this strategy captured the broadest and most relevant body of literature across databases, and that this approach was further informed by the authors’ familiarity with the fencing literature. We also now acknowledge as a potential limitation that studies referring only to specific disciplines (e.g., foil or sabre) without explicit reference to fencing in the title or abstract may theoretically have been missed. We believe these revisions strengthen transparency, reproducibility, and justification of the search strategy. (Please see lines 117-130)

Comment 4: Fourth, the data availability statement should be improved. Because this review generated a substantial extracted dataset across 445 studies, the underlying extraction sheet and coding framework should ideally be made available in a public repository or as supporting information, in line with PLOS ONE data availability expectations. The current statement that data will be available on acceptance is not fully sufficient as written.

RESPONSE: Thank you for this comment. We agree that the extracted dataset represents an important output of this scoping review. The full data extraction sheet has now been made publicly available via the following doi: 10.18745/ds.00026640. The dataset includes clearly labelled variables, and a corresponding README file provides detailed descriptions of all variables and coding categories to support transparency and reuse. The Data Availability Statement has been updated accordingly.

Comment 5: Fifth, I noticed what appears to be a presentation error in the figure captions. The captions/descriptions for Figures 2 and 3 seem to be reversed: the text under Figure 2 describes sample size distribution, while the text under Figure 3 describes annual publication counts. Please check and correct this.

RESPONSE: We thank the reviewer for identifying this presentation error. The captions for Figures 2 and 3 were inadvertently reversed in the original submission. This has been corrected in the revised manuscript so that Figure 2 now corresponds to annual publication counts and Figure 3 corresponds to sample size distribution.

Comment 6: I also encourage the authors to slightly moderate some of the wording in the Discussion. For example, the “Structural Imbalance Model” is an interesting interpretive summary, but it may be better presented as a conceptual interpretation of the mapped literature rather than as a formal model. Similarly, some claims about what the field has “prioritized” could be framed more cautiously.

RESPONSE: We thank the reviewer for this thoughtful suggestion. In response, we removed the “Structural Imbalance Model” heading and revised the text to present this as an interpretive conceptual summary within the Principal Findings section rather than as a formal model. We also moderated wording throughout the Discussion to frame statements more cautiously, including revising language suggesting the field has “prioritized” certain areas and softening related phrasing regarding emphasis, funding structures, and future research priorities. We believe these revisions improve precision and avoid overstatement while preserving the intended interpretation of the mapped literature. (Please see lines 255, 258-259, 342, 349, and 404.)

Reviewer 2

Comment 1: The author mentioned usage of AI tool in data screening and extraction, but details about the AI are insufficient and lead to less transparency and reproducibility. For example, are there any validity test done on the AI tool and what prompt was given to this tool to minimize the bias created by AI? Are there any criteria used to ensure accuracy?

RESPONSE: We thank the reviewer for this important comment. In response, we substantially expanded the Methods section to provide additional detail regarding the role of AI-assisted tools in screening, extraction, and translation, including how these tools were used, how outputs were verified, and what safeguards were in place to support consistency and accuracy. Specifically, we clarify that AI-assisted tools were used only to support identification and organization of relevant information and did not make autonomous inclusion, exclusion, or coding decisions. We added detail regarding the predefined criteria used when AI assistance was applied, including eligibility clarification related to participant inclusion, study design, and fencing-specific reporting in multi-sport studies, as well as standardized extraction fields and categorical definitions used to support consistency across studies. We also expanded the description of reviewer-led safeguards, including manual review and verification of all AI-assisted outputs, cross-checking against source manuscripts, and resolution of uncertainties or discrepancies prior to final coding.

Formal validity testing of the AI tool was not performed, and we have not presented it as such; rather, accuracy was supported through these reviewer-led verification procedures. We also added citations to recent emerging evidence supporting the use of human oversight in Elicit-assisted extraction workflows (Hilkenmeier et al., 2025) and to the screening platform used in this review (Rayyan; Ouzzani et al.). With respect to prompts, AI assistance in this review was applied through predefined eligibility criteria and standardized extraction fields, which are now described in greater detail in the revised Methods, rather than through prompt-based optimization intended to tune outputs. We believe these revisions substantially improve transparency, reproducibility, and justification of the AI-assisted methods used in this review.

(Please see lines 117-130 and 143-152)

Comment 2: Line 151: missing period.

RESPONSE: Thank you for noting this, the period has been added (please see line 173).

Comment 3: For result section the depth of data synthesis and analysis are very limited. Most analysis was descriptive (frequencies and percentages). Deeper insight with some exploration of relationships between variables is recommended (e.g., study design × domain, geography × design).

RESPONSE: We thank the reviewer for this thoughtful suggestion. Consistent with scoping review methodology, the analysis in this study was intentionally focused on descriptive mapping of the structure, characteristics, and distribution of the literature rather than inferential or exploratory analyses of relationships between study characteristics. The objective of the review was to characterize the field rather than test associations between variables. This approach is aligned with PRISMA-ScR guidance, which describes scoping reviews as intended to map evidence, identify key concepts and knowledge gaps, and summarize charted data, rather than require inferential analyses of associations (Tricco et al., 2018

Tricco AC, Lillie E, Zarin W, O’Brien KK, Colquhoun H, Levac D, et al. PRISMA extension for scoping reviews (PRISMA-ScR): checklist and explanation. Ann Intern Med. 2018;169(7):467-473. doi:10.7326/M18-0850.

---

## [Decision Letter · Decision Letter 1]

13 May 2026

Experimental and Quantitative Observational Research in Fencing and Wheelchair Fencing: A Scoping Review

PONE-D-26-11865R1

Dear Dr. Bottoms,

We’re pleased to inform you that your manuscript has been judged scientifically suitable for publication and will be formally accepted for publication once it meets all outstanding technical requirements.

Kind regards,

Yih-Kuen Jan, PhD

Academic Editor

PLOS One

Additional Editor Comments (optional):

Reviewers' comments:

Reviewer's Responses to Questions

**Comments to the Author**

1. If the authors have adequately addressed your comments raised in a previous round of review and you feel that this manuscript is now acceptable for publication, you may indicate that here to bypass the “Comments to the Author” section, enter your conflict of interest statement in the “Confidential to Editor” section, and submit your "Accept" recommendation.

Reviewer #1: All comments have been addressed

Reviewer #2: All comments have been addressed

2. Is the manuscript technically sound, and do the data support the conclusions?

Reviewer #1: Yes

Reviewer #2: Yes

3. Has the statistical analysis been performed appropriately and rigorously? 

Reviewer #1: N/A

Reviewer #2: N/A

4. Have the authors made all data underlying the findings in their manuscript fully available?

Reviewer #1: Yes

Reviewer #2: Yes

5. Is the manuscript presented in an intelligible fashion and written in standard English?

Reviewer #1: Yes

Reviewer #2: Yes

6. Review Comments to the Author

Reviewer #1: (No Response)

Reviewer #2: The authors addressed the comments made during the first review. The article focuses on descriptive mapping of literatures regarding fencing.

7. PLOS authors have the option to publish the peer review history of their article (what does this mean?). If published, this will include your full peer review and any attached files.

Reviewer #1: No

Reviewer #2: No

---

## [Editor Report · Acceptance letter]

PONE-D-26-11865R1

PLOS One

Dear Dr. Bottoms,

I'm pleased to inform you that your manuscript has been deemed suitable for publication in PLOS One. Congratulations! Your manuscript is now being handed over to our production team.

Kind regards,

on behalf of

Dr. Yih-Kuen Jan

Academic Editor

PLOS One